

Atmospheric
Measurement
Techniques

# Neural-network-based estimation of regional-scale anthropogenic CO$_2$ emissions using an Orbiting Carbon Observatory-2 (OCO-2) dataset over East and West Asia

**Farhan Mustafa[1], Lingbing Bu[1], Qin Wang[1], Na Yao[1], Muhammad Shahzaman[2], Muhammad Bilal[3], Rana Waqar Aslam[4], and Rashid Iqbal[5]**

[1]Collaborative Innovation Center on Forecast and Evaluation of Meteorological Disasters, Key Laboratory for Aerosol-Cloud-Precipitation of China Meteorological Administration, Key Laboratory of Meteorological Disasters, Ministry of Education, Nanjing University of Information Science and Technology (NUIST), Nanjing 210044, China
[2]School of Atmospheric Sciences (SAS), Nanjing University of Information Science and Technology (NUIST), Nanjing 210044, China
[3]School of Marine Sciences (SMS), Nanjing University of Information Science and Technology (NUIST), Nanjing 210044, China
[4]State Key Laboratory of Information Engineering in Surveying, Mapping and Remote Sensing (LIESMARS), Wuhan University, Wuhan 430079, China
[5]Department of Agronomy, Faculty of Agriculture and Environment, The Islamia University of Bahawalpur, Bahawalpur 63100, Pakistan

**Correspondence:** Lingbing Bu (lingbingbu@nuist.edu.cn)

**Abstract.** [CE1] Atmospheric carbon dioxide (CO$_2$) is the most significant greenhouse gas, and its concentration is continuously increasing, mainly as a consequence of anthropogenic activities. Accurate quantification of CO$_2$ is critical for addressing the global challenge of climate change and for designing mitigation strategies aimed at stabilizing CO$_2$ emissions. Satellites provide the most effective way to monitor the concentration of CO$_2$ in the atmosphere. In this study, we utilized the concentration of the column-averaged dry-air mole fraction of CO$_2$, i.e., XCO$_2$ retrieved from a CO$_2$ monitoring satellite, the Orbiting Carbon Observatory-2 (OCO-2), and the net primary productivity (NPP) provided by the Moderate Resolution Imaging Spectroradiometer (MODIS) to estimate the anthropogenic CO$_2$ emissions using the Generalized Regression Neural Network (GRNN) over East and West Asia. OCO-2 XCO$_2$, MODIS NPP, and the Open-Data Inventory for Anthropogenic Carbon dioxide (ODIAC) CO$_2$ emission datasets for a period of 5 years (2015–2019) were used in this study. The annual XCO$_2$ anomalies were calculated from the OCO-2 retrievals for each year to remove the larger background CO$_2$ concentrations and seasonal variability. The XCO$_2$ anomaly, NPP, and ODIAC emission datasets from 2015 to 2018 were then used to train the GRNN model, and, finally, the anthropogenic CO$_2$ emissions were estimated for 2019 based on the NPP and XCO$_2$ anomalies derived for the same year. The estimated and the ODIAC CO$_2$ emissions were compared, and the results showed good agreement in terms of spatial distribution. The CO$_2$ emissions were estimated separately over East and West Asia. In addition, correlations between the ODIAC emissions and XCO$_2$ anomalies were also determined separately for East and West Asia, and East Asia exhibited relatively better results. The results showed that satellite-based XCO$_2$ retrievals can be used to estimate the regional-scale anthropogenic CO$_2$ emissions, and the accuracy of the results can be enhanced by further improvement of the GRNN model with the addition of more CO$_2$ emission and concentration datasets.

**Published by Copernicus Publications on behalf of the European Geosciences Union.**

## 1 Introduction

Climate change is one of the greatest challenges to the future of Earth, and it stems from global warming, which is accelerated by anthropogenic emissions of greenhouse gases (Lamminpää et al., 2019). The major warming effects are caused by atmospheric $CO_2$ emissions, and significant amounts of these emissions are contributed by fossil fuel combustion and some industrial activities, such as the calcination of limestone during cement production (Hutchins et al., 2017). The levels of atmospheric $CO_2$ are continuously increasing (Mustafa et al., 2020), and if these levels continue to increase at the same rate, 1.5 °C of global warming will be reached between 2030 and 2052, which will cause more climate extremes (Hoegh-Guldberg et al., 2018).

Estimates of $CO_2$ emissions at national, regional, and global levels are now widely reported and have become an important element of public policy and mitigation strategies. Many countries are making efforts to reduce $CO_2$ emissions. Over the past few decades, significant work has been carried out to compile the regional and the global inventories of $CO_2$ emissions from anthropogenic activities (Olivier et al., 2005; Janssens-Maenhout et al., 2015; Gurney et al., 2009; Oda and Maksyutov, 2015). Most of the emission inventories employ bottom-up methods using available human activity data, emission factors, and corresponding technologies. The bottom-up methods incorporate energy consumption datasets along with other information, such as fuel purity and efficiency. However, it is known that such information can be subject to errors and biases, leading to considerable discrepancies and uncertainties in emission estimates, especially in the case of rapidly growing developing economies such as China and India (Guan et al., 2012; Korsbakken et al., 2016). These discrepancies can result in ~40 % to ~100 % uncertainty in emission estimations at the country and the local scales, respectively (Peylin et al., 2013; Wang et al., 2013). Moreover, defining the uncertainty in the inventory datasets is also a challenging task, and the intercomparisons of various inventories do not necessarily reveal all of the uncertainties, as different inventories sometimes use common sources of information (Konovalov et al., 2016) CE2. It is becoming increasingly important to find efficient and reliable ways of monitoring $CO_2$ reduction progress and to evaluate how well specific $CO_2$ reduction policies are working.

Satellites provide the most effective way of monitoring atmospheric $CO_2$ with great spatiotemporal resolution. Several satellites such as the Greenhouse Gases Observing Satellite (GOSAT), GOSAT-2, the Orbiting Carbon Observatory-2 (OCO-2), OCO-3, and TanSAT CE3 are orbiting the Earth and are dedicated to monitoring atmospheric $CO_2$ (Crisp, 2015; Liu et al., 2018; Matsunaga et al., 2019; Taylor et al., 2020; Bao et al., 2020; Hong et al., 2021; Yang et al., 2018). These satellites calculate the average atmospheric $CO_2$ concentration in the path of sunlight reflected by the surface using spectrometers carried onboard. OCO-2 measures the $CO_2$

optical depth with bands centered around 1.6 and 2.0 μm TS1 and determines the $O_2$ optical depth using the A-band, which is centered around 0.76 μm (Crisp et al., 2017; O'dell et al., 2012). The information from these bands is combined to calculate the column-averaged dry-air mole fraction of $CO_2$ ($XCO_2$) (Crisp et al., 2012). Several studies suggest that $XCO_2$ can be used to detect the $CO_2$ concentration induced by anthropogenic activities by removing the background concentration from the satellite $XCO_2$ retrievals (Bovensmann et al., 2010; Hakkarainen et al., 2019; Keppel-Aleks et al., 2013). The results from these studies have reported an enhancement of nearly 2 ppm over megacities and high-density urban regions in the US and China. The $XCO_2$ retrievals derived from the satellite measurements show a positive correlation with the $CO_2$ emission inventories (Hakkarainen et al., 2016; Yang et al., 2019) which implies that these space-based observations can be used to assess the anthropogenic $CO_2$ emissions by enhancing the anthropogenic $XCO_2$ concentration.

Asia is home to the world's most populous nations with the highest $CO_2$ emissions. East Asia, in particular China, significantly contributes to the global carbon budget and has accounted for ∼ 30 % of the overall growth in global $CO_2$ emissions over the past 15 years (EDGAR TS2, 2017). This increment in the $CO_2$ levels is mainly due to the rapid economic growth and anthropogenic activities (Shan et al., 1997). China has pledged to make aggressive efforts to reduce the $CO_2$ emissions per unit gross domestic product (GDP) by 60 %–65 % relative to 2005 levels, and peak carbon emissions overall, by 2030 (UNFCCC TS3, 2015). West Asia is also a region with higher rates of anthropogenic $CO_2$ emissions (Mustafa et al., 2020), and some of its countries, such as Iran, Saudi Arabia, and Turkey, are listed among the 10 largest $CO_2$ emitting nations in the world. Several studies have been carried out to estimate the $CO_2$ emissions using various machine learning techniques, but most of them do not deal with the spatial distribution. Rao (2021) estimated the $CO_2$ emissions using Support Vector Machine (SVM). Zhonghan et al. (2018) predicted the $CO_2$ flux emissions based on published data including latitude, age, potential net primary productivity (NPP), and mean depth using the Back Propagation Neural Network (BPNN) and Generalized Regression Neural Network (GRNN) models. Yang et al. (2019) estimated the anthropogenic $CO_2$ emissions using GOSAT $XCO_2$ retrievals over China, and the results showed good agreement between the estimated values and the ODIAC $CO_2$ emission dataset. In this study, we have improved the model initially developed by Yang et al. (2019) to estimate the regional-scale anthropogenic $CO_2$ emissions using OCO-2 $XCO_2$ retrievals over East and West Asia. MODIS NPP, OCO-2, and ODIAC $CO_2$ datasets were obtained for a period of 5 years from January 2015 to December 2019. $XCO_2$ anomalies were calculated from the OCO-2 retrievals for each year; the GRNN model was trained using $XCO_2$ anomalies, MODIS NPP, and ODIAC $CO_2$ emissions with 4 years

of data from 2015 to 2018; and then anthropogenic CO$_2$ emissions were estimated for the year 2019 based on 2019 NPP and XCO$_2$ anomalies. Atmospheric CO$_2$ monitoring satellites can detect and analyze the anthropogenic CO$_2$ signatures, and the satellite-based estimation of anthropogenic CO$_2$ emissions can be helpful in investigating the carbon emissions as a data-driven method, which is different from the conventional method of calculating an emission inventory. Although the estimation of anthropogenic CO$_2$ emissions using satellite datasets is a challenging task, as some other factors such as the atmospheric transport and the terrestrial ecosystem play notable roles in controlling the spatial distribution of atmospheric CO$_2$ (Cao et al., 2017), this data-driven method can still provide meaningful help with respect to quantifying anthropogenic CO$_2$ emissions that will be important for evaluating the effects of anthropogenic CO$_2$ emission reduction at regional as well as global scales.

The remainder of this paper is structured as follows: the details of the datasets and methods are provided in Sect. 2, and the results, including the estimated CO$_2$ emissions, an evaluation of these emissions, and the correlation between ODIAC CO$_2$ emissions and XCO$_2$ anomalies are discussed in Sect. 3.

## 2 Materials and methods

### 2.1 Datasets

#### 2.1.1 OCO-2 dataset

The Orbiting Carbon Observatory-2 (OCO-2) was launched by the National Aeronautics and Space Administration (NASA) on 2 July 2014 to monitor the concentration of atmospheric CO$_2$ at regional and global levels (Crisp, 2015). It carries a three-channel imaging grating spectrometer that collects high-resolution, bore-sighted spectra of reflected sunlight. Spectra are collected in the molecular oxygen A-band at 0.765 µm and the CO$_2$ bands at 1.61 and 2.06 µm (Hakkarainen et al., 2019). Information from all of these bands is combined to calculate the XCO$_2$. The spatial resolution of OCO-2 is 2.25 km × 1.29 km. More details about the instrument design, calibration approach, in-orbit performance, and measurement principles are provided in a previous study (Crisp, 2015). In this study, we used the OCO-2 Atmospheric Carbon Observations from Space (ACOS)/XCO$_2$ version 10r product that was generated using the ACOS Level 2 Full Physics (L2FP) retrieval algorithm, which used a Bayesian optimal estimation framework to derive estimates of XCO$_2$ from spectral measurements of reflected solar radiation (O'dell et al., 2012; Crisp et al., 2012). A comprehensive study on the validation of OCO-2 XCO$_2$ retrievals against the Total Carbon Column Observing Network (TCCON) CO$_2$ dataset reported an absolute median difference of less than 0.4 ppm and a root-mean-square (RMS) differ-

ence of less than 1.5 ppm between the two datasets (Wunch et al., 2017). Similar experiments have been carried out for the validation of different versions of OCO-2 XCO$_2$ products, and the results have shown that the OCO-2 dataset was consistent and reliable for atmospheric CO$_2$ monitoring (Kiel et al., 2019; O'dell et al., 2018). The quality and the quantity of the XCO$_2$ product have been improved with the developments in the ACOS FP retrieval algorithm. The latest OCO-2 XCO$_2$ product has single sounding precision of ∼ 0.8 ppm over land and ∼ 0.5 ppm over water, and RMS biases of 0.5–0.7 ppm over both land and water (ODell et al., 2021). The evolution of the ACOS L2FP retrieval algorithm from v7 to v10 is summarized in Table 1.

No major changes were made in the ACOS v9 L2FP retrieval algorithm relative to v8 except for the sampling of the meteorological prior. The trace gas absorption coefficient tables (ABSCO) were updated in various versions of the ACOS L2FP retrieval algorithms. The source of the prior meteorology was changed from the European Center for Medium-Range Weather Forecasts (ECMWF) in ACOS v7 to the NASA Goddard Modeling and Assimilation Office (GMAO) Goddard Earth Observing System (GEOS) Forward Processing – Instrument Team (FP-IT) products for v8 and v9. The aerosol prior source was changed from the GMAO Modern-Era Retrospective analysis for Research and Applications (MERRA) product in v7–9 to Goddard Earth Observing System 5 (GEOS5) FP-IT in v10. Moreover, an additional stratospheric aerosol layer was introduced in ACOS v8–10. The prior value of aerosol optical depth (AOD) for each retrieved aerosol type was lowered from 0.0375 in v7 to 0.0125 in v8–10. The CO$_2$ prior developed by the Total Carbon Column Observing Network (TCCON) team using the ggg2014 algorithm remained same in v7, v8, and v9 of the algorithm. Another major change was switching the land surface model from a purely Lambertian land surface model to a bidirectional reflectance distribution function (BRDF) model (Taylor et al., 2021).

#### 2.1.2 ODIAC dataset

ODIAC is a global fossil fuel CO$_2$ (FFCO$_2$) emission dataset with a 1 × 1 km monthly resolution over land and a 1 × 1° annual resolution for international bunkers from the year 2000 onward (Oda et al., 2018). It shares country-scale estimates with the Carbon Dioxide Information Analysis Center (CDIAC) but distributes the emissions differently within the countries and includes gridded international bunker emissions (Oda and Maksyutov, 2015). CDIAC distributes the CO$_2$ emissions based on the population density, whereas ODIAC incorporates power plant profiles and nighttime light observations for emission distribution (Wang et al., 2020). ODIAC shows better agreement with the US bottom-up inventory (Gurney et al., 2009) than CDIAC, and it is commonly used in flux inversions (Crowell et al., 2019; Lauvaux et al., 2016; Maksyutov et al., 2013; Takagi et al., 2011).

**Table 1.** Evolution of the Atmospheric Carbon Observations from Space (ACOS) Level 2 Full Physics (L2FP) retrieval algorithm (Taylor et al., 2021).

|   |                          | ACOS v7                         | ACOS v8/9                  | ACOS v10                                    |
|---|--------------------------|---------------------------------|----------------------------|---------------------------------------------|
| 1 | Spectroscopy             | ABSCO v4.2                      | ABSCO v5.0                 | ABSCO v5.1                                  |
| 2 | Meteorology prior source | ECMWF                           | GEOS5 FP-IT                | No changes                                  |
| 3 | Aerosol prior source     | MERRA monthly climatology       | No changes                 | GEOS5 FP-IT with tightened prior uncertainty |
| 4 | Retrieved aerosol types  | Water, ice, and two MERRA types | With stratospheric aerosol | No changes                                  |
| 5 | AOD prior value (per type) | 0.0375                        | 0.0125                     | No changes                                  |
| 6 | $CO_2$ prior source      | TCCON ggg2014                   | No changes                 | TCCON ggg2020                               |
| 7 | Land surface model       | Lambertian                      | BRDF                       | No changes                                  |

In this study, we used the 2020 version of ODIAC emission dataset that is freely available and can be downloaded from http://db.cger.nies.go.jp/dataset/ODIAC/TS6.

## 2.2 Methods

The estimation of anthropogenic $CO_2$ emissions includes three major steps, as shown in Fig. 1: the first step includes enhancing the $XCO_2$ concentration influenced by anthropogenic activities; the second step involves setting up the GRNN model using the $XCO_2$, NPP, and ODIAC datasets; and the final step is the validation of estimated $CO_2$ emissions against the actual ODIAC emission dataset.

The OCO-2 $XCO_2$ dataset was downloaded from the EARTHDATA platform (https://earthdata.nasa.govTS7); to ensure the reliability of the data, screening and filtering of the dataset was carried out following the instructions given in the OCO-2 Data User Guide (DUG). Each sounding that is processed using the ACOS L2FP retrieval algorithm is assigned either a "good" (0) or "bad" (1) quality flag based on screening criteria derived from comparisons with TCCON and modeled $CO_2$ fields. It is generally advised that users should use the good-quality soundings for regional- and local-scale studies because the soundings flagged as bad-quality might include biases that compromise their utility for the application. In this study, the OCO-2 $XCO_2$ retrievals were included if (i) they were flagged good (flag of 0) and (ii) the standard deviation of the good soundings for the day was less than 2 ppm. $CO_2$ has a larger background concentration and a longer atmospheric lifetime than other greenhouse gases (Hakkarainen et al., 2019). Hence, $XCO_2$ varies by nearly 2 % over the seasonal cycle and from pole to pole. In addition, $XCO_2$ variations influenced by anthropogenic activities are also smaller on the scale of satellite soundings (2–4 km²). Therefore, high precision is critical for the accurate quantification of the $XCO_2$ anomalies related to anthropogenic activities. To highlight the emission areas, $CO_2$ seasonal variability and the large background concentrations must be removed.

To highlight the areas associated with the anthropogenic $CO_2$ emission, $XCO_2$ anomalies were calculated by subtracting the daily $XCO_2$ median (daily background) from the individual $XCO_2$ observation – a method suggested by previous studies (Hakkarainen et al., 2019, 2016):

$$XCO_2 \text{ (anomaly)} = XCO_2 \text{ (individual)}$$
$$- XCO_2 \text{ (daily background)}. \quad (1)$$

This equation calculated the $XCO_2$ anomalies for each observation. Subtraction of the daily background concentration removes the seasonal variability. The space-based soundings are irregularly distributed and have spatiotemporal gaps because a large amount of the satellite observations is removed after screening for clouds and other artifacts. To deal with the spatiotemporal gaps, kriging interpolation was used, and a mapping dataset was generated with a spatial resolution of 0.5° × 0.5° (latitude × longitude) and a temporal resolution of 16 d. Finally, the mean against each grid cellCE8 was calculated for each year from 2015 to 2019. The annual mean of $XCO_2$ (anomaly) can detrend the seasonal variation (Hakkarainen et al., 2016). The annually averaged $XCO_2$ anomalies were resampled at a grid with a spatial resolution of 1° × 1° (latitude × longitude) and used along with 1° × 1° (latitude × longitude) ODIAC emission dataset to set up the GRNN model.

During the process of photosynthesis, living plants convert $CO_2$ into sugar molecules that they use for food. In the process of making food, they also release the oxygen we breathe. Plant productivity plays a crucial role in the global carbon cycle by absorbing the $CO_2$ released by anthropogenic activities. The net primary productivity (NPP) shows how much $CO_2$ is absorbed by plants during photosynthesis minus how much $CO_2$ is released during respiration. A negative NPP value means that $CO_2$ is released into the atmosphere, and a positive value represents the absorption of atmospheric $CO_2$. To improve the model results, an NPP dataset (MOD17A3HGF) provided by MODIS has also been used in this study. It provides information about annual NPP and is distributed by NASA's Land Processes Distributed Active Archive Center (LP DAAC). The NPP dataset with a spatial resolution of 500 m was downloaded from the LP DAAC website (https://lpdaac.usgs.gov/products/mod17a3hgfv006TS8). The annual NPP is derived from the sum of all 8 dTS9 Net Photosynthesis (PSN) products (MOD17A2H) from the given year. The MODIS NPP dataset was reprojected and resampled to the spatial res-

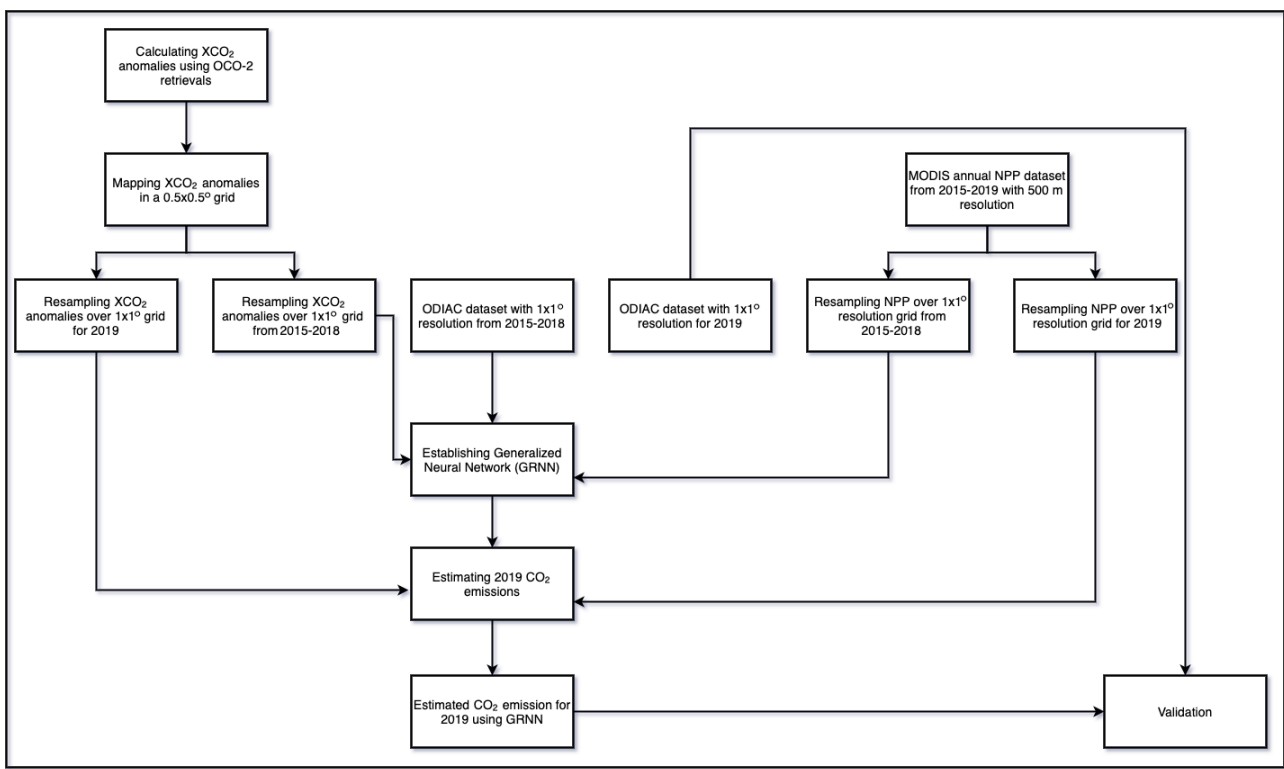

**Figure 1.** Flowchart explaining the steps involved in estimating the anthropogenic CO$_2$ emissions using MODIS NPP and OCO-2 XCO$_2$ retrievals.

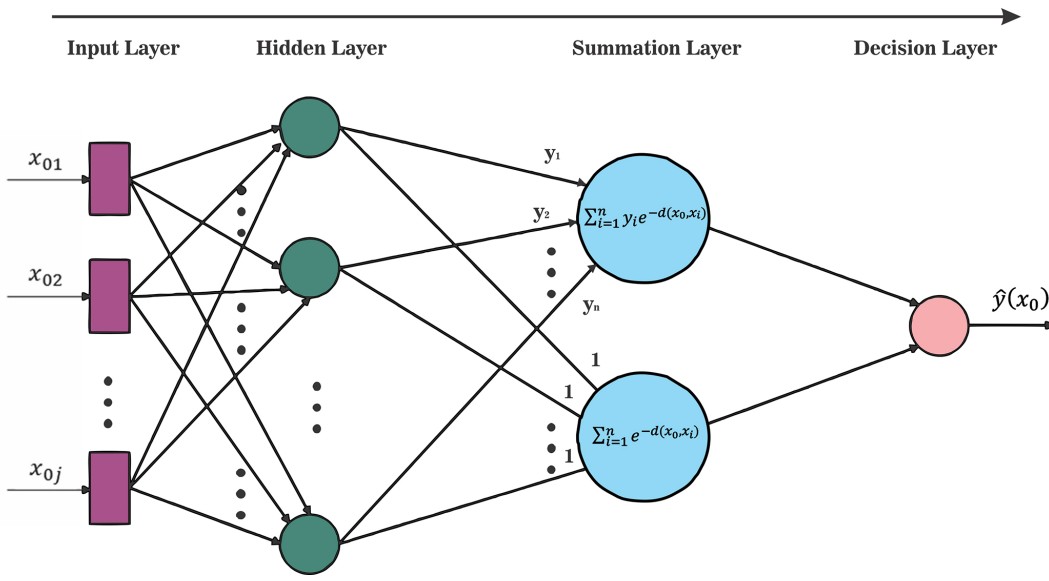

**Figure 2.** Flowchart explaining the steps involved in estimating the anthropogenic CO$_2$ emissions using OCO-2 XCO$_2$ retrievals (Yang et al., 2019).

olution of $1° \times 1°$ (latitude $\times$ longitude) for each year and used along with the ODIAC and OCO-2 datasets to train the GRNN model and as well predict the $CO_2$ emissions.

$XCO_2$ variations are primarily influenced by anthropogenic activities and terrestrial ecosystems, and there is both linear and nonlinear mapping between the $XCO_2$ and the emissions. We adopted the GRNN algorithm to represent the nonlinear mapping between the independent variables ($XCO_2$ anomaly and NPP) and the dependent variable ($CO_2$ emissions). The GRNN is a memory-based network that provides estimates of continuous variables and converges to an underlying regression. The regression of a dependent variable on an independent variable is the computation of the most probable value of the dependent variable for each value of the independent variable based on a finite number of possibly noisy measurements of the independent variable and the associated values of the dependent variable. The dependent and the independent variables are usually vectors (Rooki, 2016). The architecture of GRNN is shown in Fig. 2. It consists of four layers including an input layer, a hidden layer, a summation layer, and a decision layer. In the input layer, each neuron corresponds to the independent variable that is expressed as a mathematical function, and the independent variable values are standardized. The standardized values of the independent variable are then transferred to the neurons in the hidden layer. In this layer, each neuron stores the values of the dependent and independent variables and calculates a scalar function. The third layer, known as the summation layer, contains two neurons: the denominator summation unit, which sums the weight values being received from the hidden layer, and the numerator summation unit, which sums the weight values multiplied by the actual target-dependent variable value for each hidden neuron. Finally, the target-dependent value is obtained in the decision layer by dividing the value accumulated in the numerator summation unit by the value in the denominator summation unit. To develop a neural network, the dependent and the independent training variables must be standardized so that all training data will have the same order of magnitude in the input layer (Yang et al., 2019) CE9.

$$d(x_0 - x_i) = \sum_{j=1}^{p} \left[ \frac{x_{0j} - x_{ij}}{\sigma} \right]^2, \qquad (2)$$

where $p$ is the dimension of the variable vector $x_i$, $\sigma$ is the spread parameter, and an optimal spread parameter value is obtained after several runs following the mean squared error of the estimated values, which must be kept at a minimum (Rooki, 2016). In this study, values of spread parameters were optimized using the "Holdout Method". More details about the Holdout Method are provided in a previous study (Specht, 1991). The weight of the denominator neuron was set to 1.0. The predicted target dependent variable was

defined by the following equation:

$$\hat{y}(x_0) = \frac{\sum_{i=1}^{n} y_i e^{-d(x_0, x_i)}}{\sum_{i=1}^{n} e^{-d(x_0, x_i)}}, \qquad (3)$$

where the values calculated with the scalar function in a hidden neuron $i$ are weighted with the corresponding values of the training samples $y_i$. $n$ denotes the number of training samples.

## 3   Results and discussion

### 3.1   Spatial distribution of $XCO_2$ observations and anomalies

The satellite-based observations are sensitive to clouds and aerosols; therefore, many of the data are discarded during preprocessing due to the presence of clouds and aerosols (Mustafa et al., 2021b). Figure 3a and b show TS11 the quantity of $XCO_2$ retrievals from 2015 to 2019 on a spatial grid of $0.5° \times 0.5°$ (latitude $\times$ longitude) over West and East Asia, respectively. OCO-2 shows good spatial coverage over East Asia; however, the southern parts of the region, in particular the Tibetan Plateau, have a relatively lower number of $XCO_2$ retrievals. The Tibetan Plateau is the most extensively elevated surface on Earth, and satellite measurements show larger uncertainties over this region (Yang et al., 2019). In the case of West Asia, the southern parts of the region have a lower number of $XCO_2$ retrievals. A very large desert, the Rub' al Kahli, is located in this area; it stretches across Saudi Arabia, Yemen, Oman, and the United Arab Emirates (UAE) and often observes dust storms. The lower number of $XCO_2$ retrievals in these parts of the region might be due to the ACOS $XCO_2$ retrieval algorithm that excludes satellite measurements with a high aerosol optical depth and cloud optical thickness (Crisp et al., 2012; O'dell et al., 2012).

Figure 3c shows the spatial distribution of the 5-year averaged $XCO_2$ anomalies calculated using the method described in Sect. 2.2 over West Asia. The higher concentrations of $XCO_2$ anomalies were observed over the central parts of the region that included Iran, Kuwait, Saudi Arabia, and Iraq. Iran and Saudi Arabia are listed among the top 10 $CO_2$ emitting nations and produce over 6 % of the global $CO_2$ emissions (Jalil, 2014). In addition, Iran, Saudi Arabia, and Iraq are the major fuel consumers of the region and contribute more than 60 % of the region's total fossil fuel $CO_2$ emissions (Boden et al., 2017). Figure 4d shows the multiyear averaged $XCO_2$ anomalies over East Asia. The eastern parts of the region including eastern China, Japan, and South Korea show the highest concentrations of $XCO_2$ anomalies. China's Beijing–Tianjin–Hebei area, Korea, and Japan are the most populated urban regions with high amounts of anthropogenic emissions in the world (Mustafa et al., 2020).

Figure 3e shows the monthly averaged $XCO_2$ over East and West Asia. The monthly averaged $XCO_2$ concentrations

**Atmos. Meas. Tech., 14, 1–14, 2021**                                    **https://doi.org/10.5194/amt-14-1-2021**

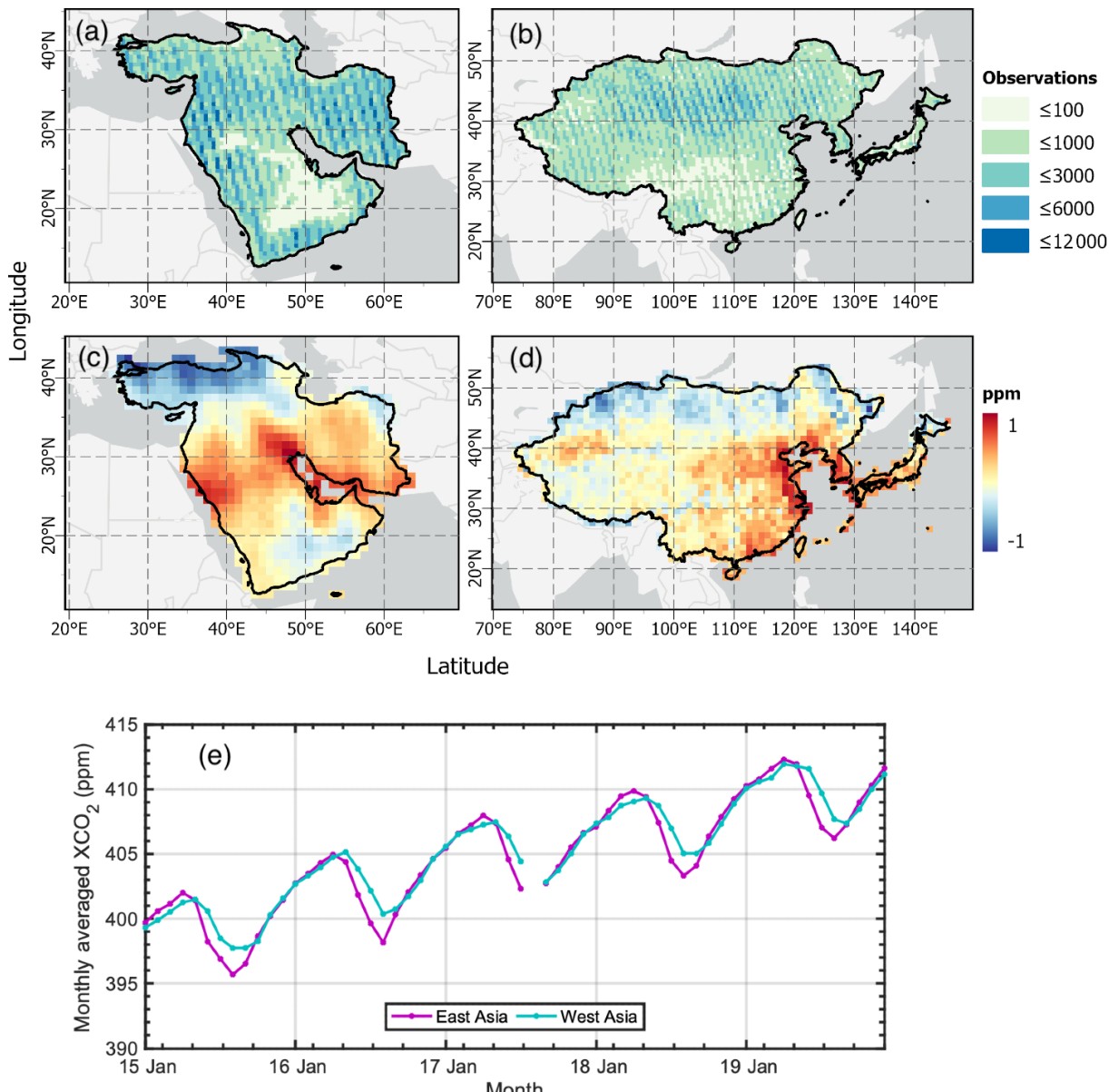

**Figure 3.** Number of observations in each cell of a 0.5 × 0.5° grid for a period of 5 years from 2015 to 2019 over **(a)** West Asia and **(b)** East Asia; the 5-year mean of XCO$_2$ anomalies calculated using OCO-2 retrievals over **(c)** West Asia and **(d)** East Asia; and **(e)** the monthly averaged XCO$_2$ concentration from 2015 to 2019 over East and West Asia. (The base map was sourced from OpenStreetMap.)`TS10`

show seasonal fluctuations. Moreover, the XCO$_2$ concentrations during each month are higher than those in the same month of the previous year, which reflects that the XCO$_2$ concentration in the atmosphere is continuously increasing in both regions. The XCO$_2$ concentration starts increasing from September and reaches its maximum value in April; it then starts decreasing and reaches its minimum value in August. The decrement in its concentration from May to August is due to several reasons; however, it is primarily owing to the strong photosynthesis and weak respiration rate of plants, which is enhanced during the monsoon or rainy season (Mustafa et al., 2020). The increment in the XCO$_2$ concentration from September to April is likely to be caused by weak photosynthesis and strong respiration, the use of heating systems in winter, and strong microbial activity (Cao et al., 2017; Mustafa et al., 2021a).

## 3.2 Estimated CO$_2$ emissions

The annually averaged XCO$_2$ anomalies, MODIS NPP, and ODIAC CO$_2$ emission datasets for a period of 4 years from 2015 to 2018 were used as a training dataset for the GRNN model built to estimate the CO$_2$ emissions using the method

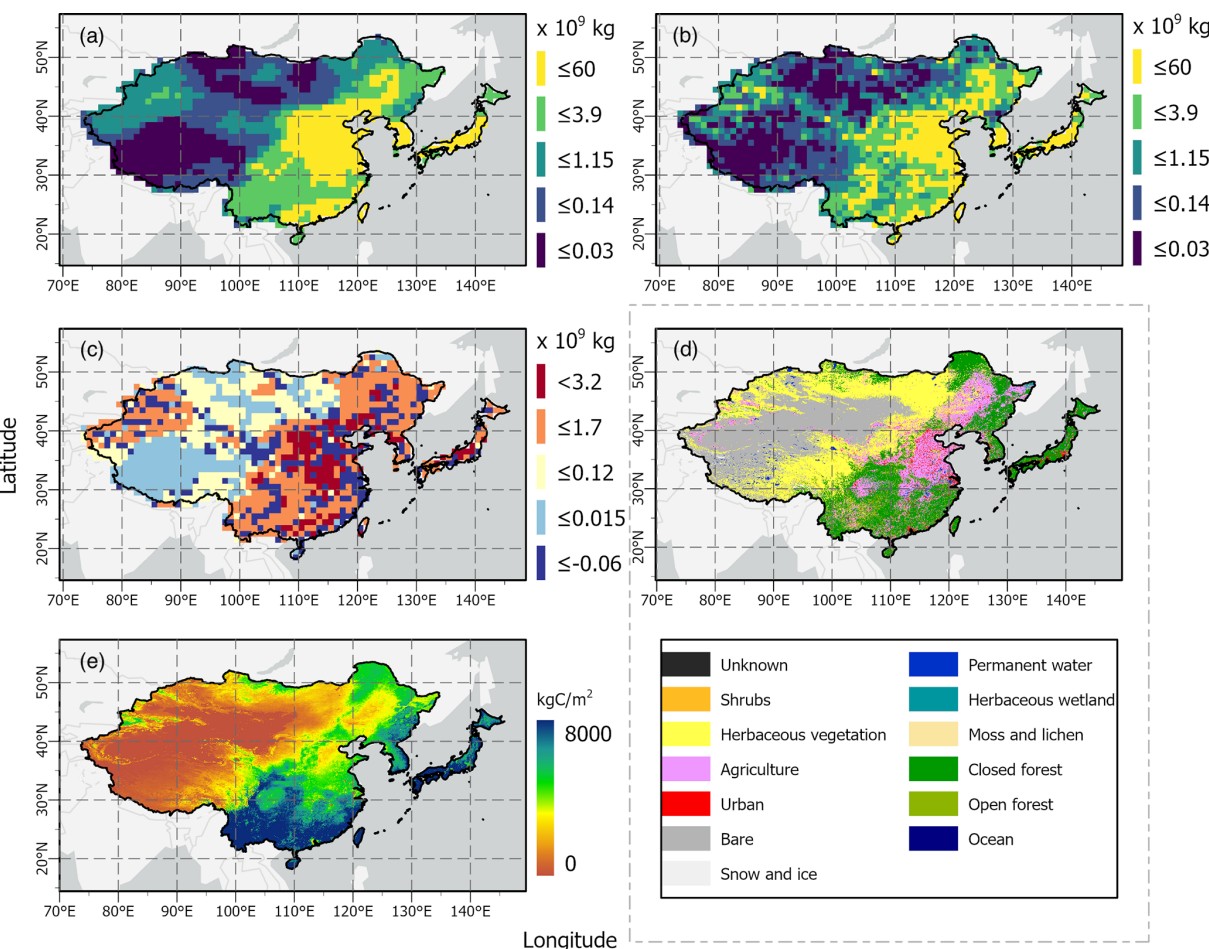

**Figure 4.** Spatial distribution of **(a)** OCO-2 $XCO_2$-based anthropogenic $CO_2$ emission estimates for 2019, **(b)** actual ODIAC emissions for 2019, **(c)** their difference (estimated emission minus CE10 actual emission), **(d)** 100 m resolution land cover distribution provided by the Copernicus Global Land Service over East Asia, and **(e)** the spatial distribution of NPP. (The base map was sourced from OpenStreetMap.)

described in Sect. 2.2. The GRNN model was then applied to 2019 annually averaged $XCO_2$ anomalies and NPP datasets to predict the $CO_2$ emissions with the same unit as the ODIAC $CO_2$ emissions. The analyses were carried out separately over East and West Asia. Figure 4a and b show TS12 the estimated values and the ODIAC $CO_2$ emissions over East Asia, respectively. The results show that the estimated values and the inventory $CO_2$ emissions exhibit nearly the same spatial distribution pattern. The eastern part of the region shows higher $CO_2$ emissions, and the western and northern parts, in particular the Tibetan Plateau and Mongolia, show the minimum $CO_2$ emissions. The pattern is also similar to the $XCO_2$ anomalies distribution over East Asia (Fig. 3d). The estimated $CO_2$ emissions have a relatively smoother distribution pattern compared with the ODIAC $CO_2$ emissions, which might be due to the interpolation of the OCO-2 dataset. Figure 4c shows the difference between the estimated and the inventory $CO_2$ emissions over East Asia. The estimated $CO_2$ emissions are generally overestimated relative to the ODIAC $CO_2$ emissions; however, the emissions are underestimated over some parts of the region as well. Figure 4d shows the land cover distribution of East Asia provided by the Copernicus Global Land Service (Buchhorn et al., 2020). The predicted $CO_2$ emissions are overestimated over most of the regional parts; however, this overestimation is more significant over agricultural areas that are located near high-density regions, e.g., eastern China. Eastern China, Japan, and Korea are known to be among the regions with the highest $CO_2$ emissions, and this underestimation over the agricultural areas might be caused by the nearby $CO_2$ emission sources which raise the $CO_2$ concentration of the nearby areas through atmospheric transport. Previous studies have demonstrated that the concentration of atmospheric $CO_2$ is influenced by atmospheric transport (Cao et al., 2017; Kumar et al., 2014). The areas where the predicted $CO_2$ emissions are underestimated are covered by agriculture, forest, and vegetation. This underestimation of the predicted $CO_2$ emissions over these areas indicates the presence of uncertainties

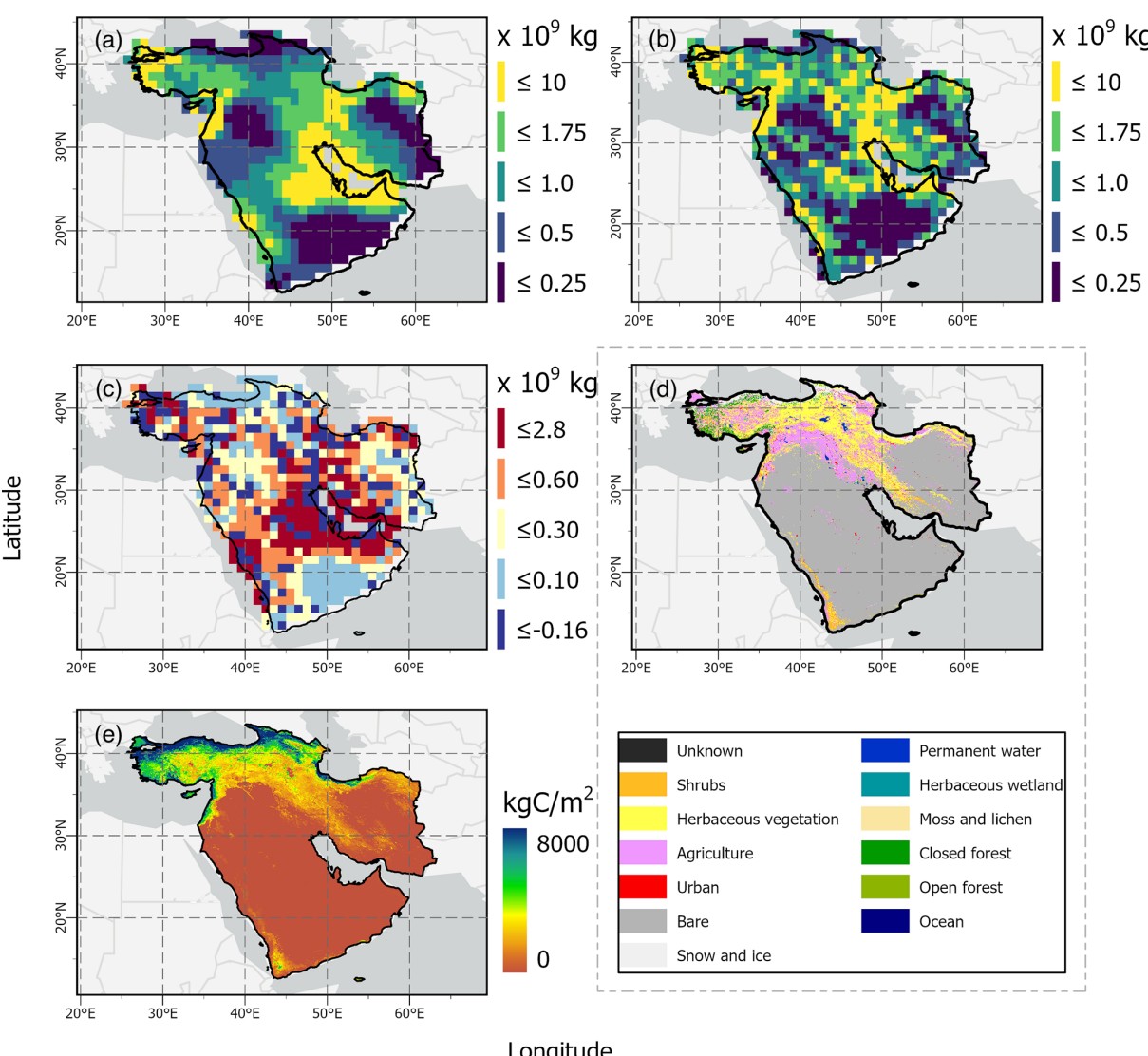

**Figure 5.** Spatial distribution of **(a)** OCO-2 $XCO_2$-based anthropogenic $CO_2$ emission estimates for 2019, **(b)** actual ODIAC emissions for 2019, **(c)** their difference (estimated emission minus actual emission), **(d)** 100 m resolution land cover distribution provided by the Copernicus Global Land Service over West Asia, and **(e)** the spatial distribution of NPP. (The base map was sourced from OpenStreetMap.)

in the $XCO_2$ anomalies that are likely to be produced by the $CO_2$ uptake of the biosphere which still remains in the $XCO_2$ anomalies. In addition, the areas where the estimated $CO_2$ emissions are overestimated have higher elevations. OCO-2 observations show larger uncertainties over elevated and mountainous areas, especially the Tibetan Plateau where the OCO-2 retrievals are significantly overestimated (Kong et al., 2019; Mustafa et al., 2020), and this might also have a contribution to the overestimation of estimated $CO_2$ emissions. The difference between the estimated and the ODIAC $CO_2$ emissions ranged from $-0.06 \times 10^9$ to $3.2 \times 10^9$ kg, and the magnitude of difference between $-1 \times 10^9$ and $1 \times 10^9$ kg accounted for 84 % of the total number of grid cells. Yang et al. (2019) estimated the $CO_2$ emissions using a similar machine learning approach with GOSAT $XCO_2$ retrievals over China, and the differences between the estimated values and the ODIAC $CO_2$ emissions were between $-5 \times 10^9$ and $5 \times 10^9$ kg. Moreover, the predicted results from the above-mentioned study exhibited less $CO_2$ emissions overall relative to the ODIAC emissions, contradicting our results. Our study showed better results, which may be due to the fact that (i) we improved the predictive model with the addition of an NPP dataset (Fig. 4e), (ii) we utilized the higher-resolution $XCO_2$ retrievals provided by OCO-2, and (iii) we incorporated the OCO-2 $XCO_2$ retrievals processed using the latest version of the retrieval algorithm. The newer version of the ACOS L2FP retrieval algorithm has improved the quan-

tity and the quality of the satellite-based observations (Taylor et al., 2021).

Figure 5a and b show TS14 the spatial distribution of satellite-based estimated $CO_2$ emissions and the actual ODIAC $CO_2$ emissions over West Asia, respectively. The spatial distribution pattern of both the estimated and the original $CO_2$ emissions is similar with some differences in their magnitudes. $CO_2$ emissions in the eastern parts are relatively larger compared with other parts of the region. Figure 5c shows the difference between the estimated values and the ODIAC $CO_2$ emissions. The satellite-based estimated $CO_2$ emissions are generally overestimated compared with the actual ODIAC $CO_2$ emissions. The estimated $CO_2$ emissions are notably larger over Iran and Saudi Arabia. Figure 5d shows the land cover distribution of West Asia. It can be seen that the predicted $CO_2$ emissions are overestimated over the areas that are covered by either urban settlements or bare land. The overestimation of estimated $CO_2$ over these areas is likely to be caused by atmospheric transportation that influences the spatial distribution of atmospheric $CO_2$ (Cao et al., 2017). Moreover, a large part of West Asia is covered by deserts, and these deserts observe a notably lower number of OCO-2 retrievals (Fig. 3a). The overestimation of the predicted $CO_2$ emissions over the largest desert of the region, the Rub' al Kahli, located in southern parts is likely to be caused by the uncertainties in the satellite-based $XCO_2$ anomalies, and these uncertainties are likely to be produced due to a lower number of OCO-2 retrievals. In addition, a previous study also indicated that the ACOS $XCO_2$ retrieval algorithm showed uncertainties over deserts (Bie et al., 2018). Similar to East Asia, the predicted $CO_2$ emissions over West Asia are also underestimated over areas that are covered by agriculture or vegetation, and this underestimation might be due to the presence of $CO_2$ uptake by the biosphere in the $XCO_2$ anomalies calculated using the satellite-based retrievals. The difference between the estimated values and the ODIAC $CO_2$ emissions ranged from $-0.16 \times 10^9$ to $2.8 \times 10^9$ kg, and the magnitude of the difference between $-1 \times 10^9$ and $1 \times 10^9$ kg accounted for 88 % of the total number of grid cell. CE12

### 3.3 Correlation analysis between OCO-2 $XCO_2$ anomalies and ODIAC emissions

Figure 6 shows the correlation analysis between the ODIAC $CO_2$ emissions and the $XCO_2$ anomalies calculated using the OCO-2 retrievals over East and West Asia. Yang et al. (2019) found that the cluster of $XCO_2$ changes derived from satellite-based observations showed a better and more significant correlation with the $CO_2$ emissions relative to a single grid of $XCO_2$, which might have been due to the fact that the atmospheric $CO_2$ measurement is an instantaneous snapshot of the realistic atmosphere (Liu et al., 2015) CE13. For the correlation analysis, we segmented the ODIAC emissions, which were binned every $0.3 \, \mathrm{t \, yr^{-1}}$ of lgE using mean

emissions calculated from annual emissions during 2015–2019, and then carried out an analysis between the mean of the emissions and the mean of the $XCO_2$ anomalies within the binned regions. The results showed a positive and significant correlation between the two datasets. Figure 6a and b show TS15 the spatial distribution of segmented ODIAC emissions over East Asia and the scatterplot between the mean of the emissions and the mean of the $XCO_2$ anomalies, respectively. The two datasets show a positive and significant correlation with a determined coefficient ($R^2$) of 0.81. The spatial distribution of segmented ODIAC emissions over West Asia and the scatterplot between the mean of the emissions and the mean of the $XCO_2$ anomalies for this region are shown in Fig. 6c and d, respectively. The two datasets showed a good correlation with a determined coefficient ($R^2$) of 0.60. Several studies have correlated satellite-based $XCO_2$ anomalies with $CO_2$ emissions (Fu et al., 2019; Shekhar et al., 2020). Yang et al. (2019) performed a correlation analysis between the GOSAT-based $XCO_2$ anomalies and the ODIAC $CO_2$ emissions over China and found a significant correlation with a determined coefficient ($R^2$) of 0.82 which increased up to 0.95 if the analysis was carried out with higher $CO_2$ emission values. In our study, the correlation between the $CO_2$ emissions and $XCO_2$ anomalies is relatively low for West Asia, which might be due to the uncertainties in the OCO-2 retrievals. A large part of West Asia is covered by deserts, and, as previously stated, Bie et al. (2018) reported that the ACOS $XCO_2$ retrieval algorithm showed uncertainties over deserts.

### 4 Summary and conclusions

In this study, anthropogenic $CO_2$ emissions were estimated using satellite datasets and employing a neural-network-based method. The study was carried out using ODIAC $CO_2$ emissions, OCO-2 $XCO_2$, and MODIS NPP datasets from 2015 to 2019. To remove the $CO_2$ seasonal variability and the large background concentration from the OCO-2 $XCO_2$ retrievals, $XCO_2$ anomalies were calculated for each year. A GRNN model was then built; $XCO_2$ anomalies, NPP, and $CO_2$ emissions from 2015 to 2018 were used as a training dataset; and, finally, $CO_2$ emissions were predicted for 2019 based on the NPP and $XCO_2$ anomalies calculated for the same year. The analyses were carried out separately over East and West Asia. The satellite-based estimated values and the ODIAC $CO_2$ emission datasets were compared, and both of the datasets showed good agreement in terms of spatial distribution. The estimated $CO_2$ emissions showed better results over East Asia compared with West Asia, which might be due to the uncertainties in the $XCO_2$ retrievals: previous studies have reported that the ACOS $XCO_2$ retrieval algorithm produced uncertainties over deserts. The predicted $CO_2$ emissions were generally overestimated, and this overestimation was larger over the areas that were closer to the high-density urban regions. The overestimations might be

**Atmos. Meas. Tech., 14, 1–14, 2021** **https://doi.org/10.5194/amt-14-1-2021**

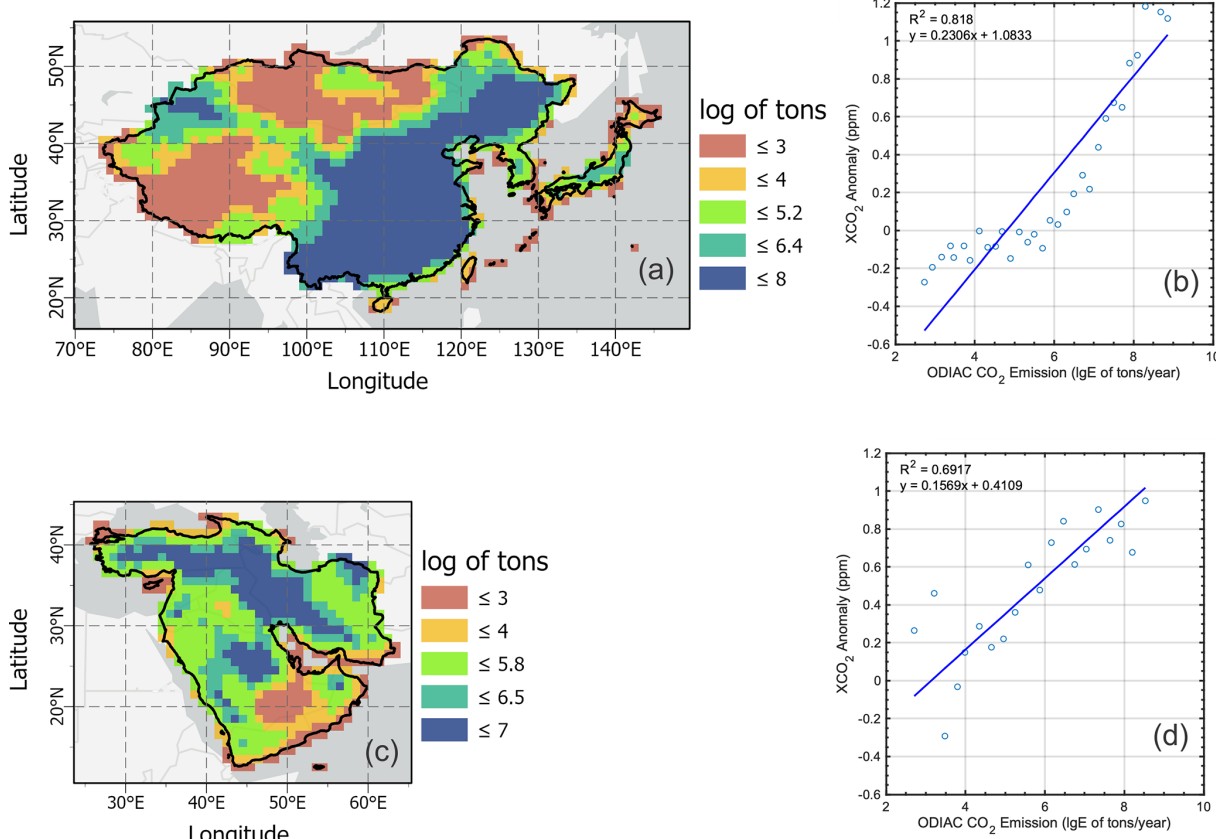

**Figure 6.** The spatial distribution of segmented ODIAC emissions, where the data are binned in 0.3 t yr$^{-1}$ TS13 of lgE CE11 bins using the mean emission calculated from the annual emissions from 2015 to 2019 over **(a)** East Asia and **(c)** West Asia. The correlation between mean ODIAC CO$_2$ emissions and mean XCO$_2$ anomalies calculated from annual XCO$_2$ from 2015 to 2018 for **(b)** East Asia and **(d)** West Asia. (The base map was sourced from OpenStreetMap.)

due to the nearby high-emission CO$_2$ sources that raised the XCO$_2$ concentration due to the effects of atmospheric transport. The satellite-based estimated CO$_2$ emissions were underestimated over some parts of the regions, mostly areas covered by agricultural land and vegetation; this was likely caused by the uncertainties in the calculated XCO$_2$ anomalies, and these uncertainties were produced due to the presence of the CO$_2$ uptake of the biosphere. We compared our results with a previous study carried out using a similar predictive model incorporating GOSAT XCO$_2$ retrievals CE14. The referenced study generally underestimated the predicted CO$_2$ emissions, with larger differences relative to ODIAC CO$_2$ emissions, contradicting our results. Our study showed relatively better results, which might be due to several reasons: (i) we improved the predictive model with the addition of an NPP dataset, (ii) we incorporated OCO-2 XCO$_2$ retrievals that have a higher spatial resolution compared with the GOSAT XCO$_2$ retrievals, and (iii) we used a XCO$_2$ product processed using the latest version of the ACOS L2FP retrieval algorithm. The newer version of the algorithm has improved the quantity and the quality of the XCO$_2$ retrievals.

Moreover, correlation analysis was also carried out between the ODIAC CO$_2$ emissions and the OCO-2 XCO$_2$ anomalies, and the results were significant with $R^2$ values of 0.81 and 0.60 over East and West Asia, respectively. These results were in agreement with the previous studies.

The results from our study suggest that CO$_2$ emissions can be estimated using observations obtained from CO$_2$ monitoring satellites. Currently, several satellites are orbiting the Earth and are dedicated to monitoring atmospheric CO$_2$. Joint utilization of the observations from the old and the latest satellites, such as OCO-3, GOSAT-2, and TanSAT, might reduce the spatiotemporal gaps and uncertainties. In future studies, we intend to improve the GRNN model via the addition of CO$_2$ uptake datasets and the joint utilization of multisensor data.

*Data availability.* The OCO-2 Level 2 XCO$_2$ product is available from https://earthdata.nasa.gov TS16, and the ODIAC CO$_2$ emission dataset is available from http://db.cger.nies.go.jp/dataset/ODIAC/ (CGER, 2021) TS17.

*Author contributions.* FM carried out the analysis under the supervision of LB, with input and support from QW, NY, MS, MB, RWA, and RI. FM wrote the original article with feedback from all the co-authors.

*Competing interests.* The contact author has declared that neither they nor their co-authors have any competing interests.

*Acknowledgements.* The authors acknowledge the efforts of NASA with respect to providing the OCO-2 data products. The authors are also thankful to the National Institute of Environmental Studies (NIES) for providing the ODIAC $CO_2$ emission dataset. The lead author (Farhan Mustafa) is thankful to Thomas E. Taylor (Colorado State University, USA) for providing help with summarizing the evolution of the ACOS L2FP retrieval algorithm.

*Financial support.* This research has been supported by the National Natural Science Foundation of China (grant no. 41675133). TS18

*Review statement.* This paper was edited by Dmitry Efremenko and reviewed by two anonymous referees.

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

## Remarks from the language copy-editor

CE1    Please note that this manuscript has undergone copy-editing according to the standards of American English.

CE2    Please check that the meaning of your sentence is intact.

CE3    Please note that, according to our standards, we always define abbreviations/acronyms at the first instance of use (in the abstract as well as in the rest of the text). Please check the other edits made in such cases throughout the paper and let me know if further changes are required. Thank you.

CE4    Please confirm the change.

CE5    Do you mean "single-sounding precision"?

CE6    Do you mean "1 km $\times$ 1 km"?

CE7    Do you mean "1° $\times$ 1°"?

CE8    Do you mean "the mean of each grid cell"?

CE9    Please check that the meaning of your sentence is intact.

CE10    Please confirm the change here and again in the caption of Fig. 5.

CE11    What does "lgE" refer to? This is currently unclear. Please advise.

CE12    This seems to be repeated information from page 8. Is is required again here? Please check.

CE13    Please check that the meaning of your sentence is intact.

CE14    Please add a citation for the study that you are referring to here.

## Remarks from the typesetter

TS1    Please check.

TS2    Please check and confirm the change.

TS3    Please check and confirm the change.

TS4    Please confirm or provide a different short running title.

TS5    Reference missing from reference list.

TS6    Please provide date of last access.

TS7    Please provide date of last access.

TS8    Please provide date of last access.

TS9    Please confirm.

TS10    The composition of Fig. 3 has been adjusted to our standards. Please also not the language edits to Figs. 1 and 3.

TS11    Please check.

TS12    Please check.

TS13    Please note that units have been changed to exponential format throughout the text. Please check all instances.

TS14    Please check.

TS15    Please check.

TS16    Please provide a direct link to the dataset and, if possible, a DOI instead of a URL. In any case, please provide a reference list entry including creators, title, and date of last access.

TS17    Please confirm citation.

TS18    Please note that the funding information has been added to this paper. Please check if it is correct. Please also double-check your acknowledgements to see whether repeated information can be removed or changed accordingly. Thanks.

TS19    Please ensure that any datasets and software codes used in this work are properly cited in the text and included in this reference list. Thereby, please keep our reference style in mind, including creators, titles, publisher/repository, persistent identifier, and publication year. Regarding the publisher/repository, please add "[dataset]" or "[code]" to the entry (e.g. Zenodo [code]).

TS20    Please provide article number or page range.

TS21    Please check and confirm addition of all necessary information taken from the original document.

TS22    Please provide date of last access and confirm the year. Please also confirm the entire reference list entry.

TS23    Please check and confirm the change of all author names and initials based on the original document/DOI.

TS24    Please add the title and venue of the event.

TS25    Please check and confirm the change.

TS26    Please provide article number or page range.

TS27    Please add the name of publisher/publishing institution and place of publication.

TS28    Please check and confirm the change/addition

TS29 Please provide article number or page range.
TS30 Please add the name of publisher/publishing institution and place of publication.
TS31 Reference not cited in text.
TS32 Please add the place of publication.
TS33 Please add additional data to adequately locate the source.
TS34 Please add the name of publisher/publishing institution and place of publication.
TS35 Please check and confirm the change.
TS36 Please add the name of publisher/publishing institution and place of publication.