# Peer review of "Neural Network Based Estimation of Regional Scale Anthropogenic CO2 Emissions Using OCO-2 Dataset Over East and West Asia"

_Atmospheric Measurement Techniques, 2021_

## Referee Comment (RC1)

Anonymous Reviewer

August 30, 2021

**1   General comments**

In this study, the authors proposed a method to estimate the regional scale anthropogenic $CO_2$ emissions with OCO-2 $XCO_2$ retrievals over East and West Asia. The topic fits well to the aims and scopes of AMT. Concerning critical requirement for quantitative estimates of carbon emissions and the rapid development of machine learning techniques, this study would be certainly interesting to the community. However, the current version of the manuscript, in my opinion, cannot be recommended for publication. I do have some major concerns that need to be responded if the authors consider to submit the revised manuscript.

First of all, I see little scientific significance in this paper, actually after I read the paper by Yang et al. (2019) `https://www.mdpi.com/1424-8220/19/5/1118`, I surprisingly found out there are many similarities in both papers, even though the old one has been cited by the authors. For example, Section 2.2 is quite similar to Section 2.3 on Yang et al. (2019), including all equations and Figures 1 and 2. I am not saying that the methodology (algorithms, processing steps) should not be reused especially when its performance has been justified in previous studies. But the authors claimed in the manuscript that "we proposed a method to estimate the regional scale anthropogenic $CO_2$ emissions", which can be misleading to readers. The only differences between these two papers seem to be that Mustafa et al. used OCO-2 data and extended the study region to West Asia. Therefore, if possible, I would suggest the authors to highlight the differences in both estimation methods, if not, please completely revise the manuscript for readers to better understand the objective of this paper.

Second, significant technical details are missing in Sections 2 and 3:

- Although this study directly used the $XCO_2$ product, it would be important for readers to know essential information of the retrieval algorithm, as the authors claimed in Section 3.2 that compared to previous studies, this study obtained a better result

due to the improvements in the $XCO_2$ retrieval algorithm. Therefore, in Section 2.1.1, please add more relevant details.

- Information about training, testing, validation of the GRNN should be given, e.g., what are input parameters, only OCO-2 data? How did you organize the training, testing, and validation datasets?

**2  Specific comments**

- Line 46: Seriously, I don't think your own paper is proper for this statement "Satellites provide the most effective way to monitor atmospheric $CO_2$ with great spatiotemporal resolutions". Satellite remote sensing has been utilized to measure greenhouse gases for over 20 years, and it is widely known that this technology can provide high-resolution $CO_2$ observations.

- Line 48: The references for satellite $CO_2$ measurements can be largely improved. For instance, there have been a number of new studies available for TanSAT $CO_2$ retrievals, which cannot be simply overlooked, e.g., Bao et al. (2020); Yang et al. (2018); Hong et al. (2021). In addition, it would be nicer to have journal papers instead of a conference abstract.

- Section 2.1.1: What is the spatial resolution of OCO-2? How good is the data quality of the employed $XCO_2$ retrieval product?

- Section 2.1.2: Where do you acquire ODIAC dataset? Is it publicly available? Please specify it.

- Line 216: Both "tons" and "Mt" are not SI base or SI-accepted units. Please check information at `https://www.bipm.org/documents/20126/41483022/SI-Brochure-9-EN.pdf/2d2b50bf-f2b4-9661-f402-5f9d66e4b507?version=1.9&download=true`.

- Line 217: What are the "improvements in the $XCO_2$ retrieval algorithm"? Again, does this sentence just prove that this study USED the method proposed in (Yang et al., 2019), but with a different dataset?

- Page 9–13: Many references do not have the standard format, journal names are missing in many cases.

- Figure 3: Please correct the subfigure index in the caption.

**References**

Z. Bao, X. Zhang, T. Yue, L. Zhang, Z. Wang, Y. Jiao, W. Bai, and X. Meng. Retrieval and validation of XCO2 from TanSat target mode observations in Beijing. *Remote Sens.*, 12(18), 2020. doi: 10.3390/rs12183063. 3063.

X. Hong, P. Zhang, Y. Bi, C. Liu, Y. Sun, W. Wang, Z. Chen, H. Yin, C. Zhang, Y. Tian, and J. Liu. Retrieval of global carbon dioxide from TanSat satellite and comprehensive validation with TCCON measurements and satellite observations. *IEEE Trans. Geosci. Remote Sens.*, pages 1–16, 2021. doi: 10.1109/TGRS.2021.3066623.

D. Yang, Y. Liu, Z. Cai, X. Chen, L. Yao, and D. Lu. First global carbon dioxide maps produced from TanSat measurements. *Adv. Atmos. Sci.*, 35(6):621–623, 2018. doi: 10.1007/s00376-018-7312-6.

S. Yang, L. Lei, Z. Zeng, Z. He, and H. Zhong. An assessment of anthropogenic $CO_2$ emissions by satellite-based observations in China. *Sensors*, 19(5), 2019. doi: 10.3390/s19051118. 1118.

---

## Author Comment (AC1)

**Replies to Reviewers' comments**

**Ms. Ref. No. : amt-2021-222**

**Title: Neural Network Based Estimation of Regional Scale Anthropogenic CO2 Emissions Using OCO-2 Dataset Over East and West**

We sincerely thank the Editor of the journal for reviewing our research paper and providing the list of comments/suggestions raised by the learned reviewers which in turn helped us in improving the quality of an earlier version of the manuscript. As per the suggestions of the reviewers, we have gone through the entire paper giving suitable answers to their queries and revised the whole paper. We have updated the figures following the suggestions of the reviewers. The authors wish to thank the Editor of the journal for his encouragement and support in contacting the reviewers to complete the peer-review process in time. The authors are also grateful to the anonymous reviewers for their constructive and useful comments, suggestions and critics which in turn improved the scientific content of an earlier version of the manuscript. All responses to the reviewers' comments in the revised manuscript are highlighted in RED, so that they may be easily identified.

Kind regards, Farhan Mustafa & Co-authors

**Response to Anonymous Referee 1 Comments**

**1. General Comments**

Point 1: In this study, the authors proposed a method to estimate the regional scale anthropogenic CO2 emissions with OCO-2 XCO2 retrievals over East and West Asia. The topic fits well to the aims and scopes of AMT. Concerning critical requirement for quantitative estimates of carbon emissions and the rapid development of machine learning techniques, this study would be certainly interesting to the community. However, the current version of the manuscript, in my opinion, cannot be recommended for publication. I do have some major concerns that need to be responded if the authors consider to submit the revised manuscript. First of all, I see little scientific significance in this paper, actually after I read the paper by Yang et al. (2019) https://www.mdpi.com/1424-8220/19/5/1118, I surprisingly found out there are many similarities in both papers, even though the old one has been cited by the authors. For example, Section 2.2 is quite similar to Section 2.3 on Yang et al. (2019), including all equations and Figures 1 and 2. I am not saying that the methodology (algorithms, processing steps) should not be reused especially when its performance has been justified in previous studies. But the authors claimed in the manuscript that we proposed a method to estimate the regional scale anthropogenic CO2 emissions, which can be misleading to readers. The only differences between these two papers seem to be that Mustafa et al. used OCO-2 data and extended the study region to West Asia. Therefore, if possible, I would suggest the authors to highlight the differences in both estimation methods, if not, please completely revise the manuscript for readers to better understand the objective of this paper.

**Response 1:** We are thankful to the anonymous referee for his/her constructive comments. The comments are very helpful in improving the quality of the manuscript and we have carefully used them to revise the manuscript.

We understand the concerns of the learned referee about the similarities between our manuscript and the article authored by (Yang et al., 2019). We extended the study following the suggestion given in the conclusion of the article written by Yang et al. (2019). However, following the suggestion of the respected reviewer, the manuscript has been revised completely and substaintial changes have been made in the revised version of the manuscript.

• The prediction model has been changed/improved. A new dataset, MODIS net primary productivity (NPP) has been added to train the model and then predict the anthropogenic CO2 emission. The new flowchart of the model, updated in the revised manuscript as, "Figure 1" is given in the following:

• More detail has been added to the section 2.2 of the revised manuscript at

L196-203 as, "OCO-2 XCO2 dataset was downloaded from the Earthdata platform (https://earthdata.nasa.gov/) and to ensure the reliability of the data, screening and filtering of the dataset was carried out following the instructions given in the OCO-2 Data User Guide (DUG). Each sounding that is processed using the ACOS L2FP retrieval algorithm is assigned either a "good" (=0) or "bad" (=1) quality flag based on screening criteria derived from comparisons with TCCON and modelled CO2 fields. It is generally advised that users should use the "good" quality soundings for regional and local scale studies because the soundings flagged as "bad" quality might include biases that compromise their utility for the application. In this study, the OCO-2 XCO2 retrievals were included if: (i) they were flagged good (flag=0) and (ii) the standard deviation of the good soundings for the day was less than 2 ppm."

L245-256 as, "During the process of photosynthesis, the living plants convert the  $CO_2$  into sugar molecules they use for food. In the process of making food, they also release the oxygen we breathe. Plant productivity plays a crucial role in the global carbon cycle by absorbing the CO2 released by anthropogenic activities. The net primary productivity (NPP) shows how much  $CO_2$  is absorbed by the plants during photosynthesis minus how much  $CO_2$  is released during respiration. A negative value of NPP means that  $CO_2$  is released into the atmosphere and a positive value represents the absorption of atmospheric CO2. To improve the model results, an NPP dataset (MOD17A3HGF) provided by MODIS has also been used in this study. It provides information about annual NPP and is distributed by NASA's Land Processes Distributed Active Archive Center (LP DAAC). The NPP dataset with a spatial resolution of 500 meters (m) was downloaded from the LP DAAC website (https://lpdaac.usgs.gov/products/mod17a3hgfv006/). The annual NPP is derived from the sum of all 8-day Net Photosynthesis (PSN) products (MOD17A2H) from the given year. The MODIS NPP dataset was reprojected and resampled to the spatial resolution of  $1^{\circ} \times 1^{\circ}$ Longitude/Latitude for each year and used along with the ODIAC and OCO-2 datasets to train the GRNN model and as well predicting the CO2 emission."

• The sentence including, "we proposed a new method has been revised" and the author Yang et al., (2019) is given proper credit at various places of the manuscript. Such as,

At L105-107 as, "In this study, we have improved the model initially developed by (Yang et al., 2019) to estimate the regional scale anthropogenic  $CO_2$  emissions using OCO-2 XCO2 retrievals over East and West Asia. MODIS NPP, OCO-2 and ODIAC CO2 datasets were obtained for a period of five years from January 2015 to December 2019."

At L381-388 as, "(Yang et al., 2019) estimated the CO2 emissions by a similar machine learning approach using GOSAT XCO2 retrievals over China and differences between the estimated and the ODIAC actual CO2 emissions were between  $-5x10^9$  kg to  $5x10^9$  kg. Moreover, the predicted results from the referenced study exhibited overall less CO2 emissions relative to the ODIAC emissions contradicting our results. Our study showed better results and it might be due to several reasons; (i) we improved the prediction model with the addition of NPP dataset (Figure 4e), (ii) we utilized the higher resolution XCO2 retrievals provided by OCO-2, and (iii) we incorporated the OCO-2 XCO2 retrievals processed using the latest version of the retrieval algorithm. The newer version of the ACOS L2FP retrieval algorithm has improved the quantity as well as the quality of the satellite-based observations (Taylor et al., 2021)."

- Figure 1 (given above) has been changed as the model has been changed with the addition of new dataset.
- Figure 2 is a general structure of GRNN, however, the figure has been properly cited.

**Point 2:** Although this study directly used the XCO2 product, it would be important for readers to know essential information of the retrieval algorithm, as the authors claimed in Section 3.2 that compared to previous studies, this study obtained a better result due to the improvements in the XCO2 retrieval algorithm. Therefore, in Section 2.1.1, please add more relevant details

**Response 2:** We are thankful to the reviewer for valuable suggestion. The updates in the XCO2 retrieval algorithm have summarized in the revised version of the manuscript at:

|   |                               | ACOS v7                        | ACOS v8/9                  | ACOS v10                                           |
|---|-------------------------------|--------------------------------|----------------------------|----------------------------------------------------|
| 1 | Spectroscopy                  | ABSCO v4.2                     | ABSCO v5.0                 | ABSCO v5.1                                         |
| 2 | Meteorology prior source      | ECMWF                          | GEOS5 FP-IT                | No changes                                         |
| 3 | Aerosol prior
source       | MERRA monthly climatology      | No changes                 | GEOS5 FP-IT
with tightened
prior uncertainty |
| 4 | Retrieved aerosol types       | Water + ice + 2 MERRA
types | + stratospheric
aerosol | No changes                                         |
| 5 | AOD prior value
(per type) | 0.0375                         | 0.0125                     | No changes                                         |
| 6 | CO 2 prior source  | TCCON ggg2014                  | No changes                 | TCCON
ggg2020                                   |
| 7 | Land surface model            | Lambertian                     | BRDF                       | No changes                                         |

Table 1: Evolution of ACOS L2FP retrieval algorithm (Taylor et al., 2021).

At L150-164 as, "The quality and the quantity of the  $XCO_2$  product have been improved with the developments in the ACOS FP retrieval algorithm. The evolution of the ACOS L2FP retrieval algorithm from v7 to v10 is summarized in Table 1.

No major changes were made in the ACOS v9 L2FP retrieval algorithm relative to v8 except for sampling of meteorological prior. The trace gas absorption coefficient tables (ABSCO) were updated in various versions of the ACOS L2FP retrieval algorithms. The source of the prior meteorology was changed from the European Center for Medium-range Weather Forecast (ECMWF) in ACOS v7 to the NASA Goddard Modeling and Assimilation Office (GMAO) Goddard Earth Observing System (GEOS) Forward Processing – Instrument Team (FP-IT) products for v8/9. The aerosol prior source was changed from the GMAO Modern-Era Retrospective analysis for Research and Applications (MERRA) product in v7-9 to GEOS5 FP-IT in v10. Moreover, an additional stratospheric aerosol layer was introduced in ACOS v8-10. The prior value of aerosol optical depth for each retrieved aerosol type was lowered from 0.0375 in v7 to 0.0125 in v8-10. The CO2 prior developed by the TCCON team using the ggg2014 algorithm remained same throughout various versions of the algorithm. Another major change was switching the land surface model from a purely Lambertian land surface model to Bi-Directional Reflectance Distribution Function (BRDF) model (Taylor et al., 2021)."

**Point 3:** Information about training, testing, validation of the GRNN should be given, e.g., what are input parameters, only OCO-2 data? How did you organize the training, testing, and validation datasets?

**Response 3:** We are thankful to the learned reviewer for valuable suggestion. The required information has been added to the revised version of the manuscript:

At L196-203 as, "OCO-2 XCO2 dataset was downloaded from the Earthdata platform (https://earthdata.nasa.gov/) and to ensure the reliability of the data, screening and filtering of the dataset was carried out following the instructions given in the OCO-2 Data User Guide (DUG). Each sounding that is processed using the ACOS L2FP retrieval algorithm is assigned either a "good" (=0) or "bad" (=1) quality flag based on screening criteria derived from comparisons with TCCON and modelled CO2 fields. It is generally advised that users should use the "good" quality soundings for regional and local scale studies because the soundings flagged as "bad" quality might include biases that compromise their utility for the application. In this study, the OCO-2 XCO2 retrievals were included if: (i) they were flagged good (flag=0) and (ii) the standard deviation of the good soundings for the day was less than 2 ppm."

At L238-242 as, "The space-based soundings are irregularly distributed and have spatiotemporal gaps because a large amount of the satellite observations is removed after screening for clouds and other artifacts. To deal with the spatiotemporal gaps, kriging interpolation was used and a mapping dataset was generated with the spatial resolution of  $0.5^{\circ} \times 0.5^{\circ}$  Longitude/Latitude and temporal resolution of 16 days. Finally, the mean against each grid cell was calculated for each year from 2015 to 2019."

**2. Specific Comments**

Line 46: Seriously, I don't think your own paper is proper for this statement "Satellites provide the most effective way to monitor atmospheric CO2 with great spatiotemporal resolutions". Satellite remote sensing has been utilized to measure greenhouse gases for over 20 years, and it is widely known that this technology can provide high-resolution CO2 observations..

**Response :** We are thankful to the reviewer for constructive comment. The citation of the paper has been removed.

Line 48: The references for satellite CO2 measurements can be largely improved. For instance, there have been a number of new studies available for TanSAT CO2 retrievals, which cannot be simply overlooked, e.g., Bao et al. (2020); Yang et al. (2018); Hong et al. (2021). In addition, it would be nicer to have journal papers instead of a conference abstract.

**Response :** We are thankful to the reviewer valuable suggestion. The references have been improved following the given suggestion.

Section 2.1.1: What is the spatial resolution of OCO-2? How good is the data quality of the employed XCO2 retrieval product?

**Response :** The required information has been added in the revised manuscript:

At L141-142 as, "The spatial resolution of OCO-2 is 2.25 km x 1.29 km."

At L151-152 as, "The latest OCO-2 XCO2 product has single sounding precision of  $\sim$ 0.8 ppm over land and  $\sim$ 0.5 ppm over water, and RMS biases of 0.5-0.7 ppm over both land and water (ODell et al., 2021)."

Section 2.1.2: Where do you acquire ODIAC dataset? Is it publicly available? Please specify it.

**Response :** We are thankful to the reviewer valuable suggestion. The detail has been added in the revised manuscript:

At L189-190 as, "In this study, we used the 2020 version of ODIAC emission dataset that is freely available and can be downloaded from http://db.cger.nies.go.jp/dataset/ODIAC/."

Line 216: Both tons and Mt are not SI base or SI-accepted units. Please check information at https://www.bipm.org/documents/20126/41483022/SI-Brochure-9-EN.pdf/2d2b50bf-f2b4-9661-f402-5f9d66e4b507?version=1.9&download=true.

**Response :** We have changed the units in figures as well as in the main text from tons to the SI unit kg.

---

## Author Comment (AC2)

**Replies to Reviewers' comments**

**Ms. Ref. No. :** amt-2021-222
**Title: Neural Network Based Estimation of Regional Scale Anthropogenic CO$_2$ Emissions Using OCO-2 Dataset Over East and West**

We sincerely thank the Editor of the journal for reviewing our research paper and providing the list of comments/suggestions raised by the learned reviewers which in turn helped us in improving the quality of an earlier version of the manuscript. As per the suggestions of the reviewers, we have gone through the entire paper giving suitable answers to their queries and revised the whole paper. We have updated the figures following the suggestions of the reviewers. The authors wish to thank the Editor of the journal for his encouragement and support in contacting the reviewers to complete the peer-review process in time. The authors are also grateful to the anonymous reviewers for their constructive and useful comments, suggestions and critics which in turn improved the scientific content of an earlier version of the manuscript. All responses to the reviewers' comments in the revised manuscript are highlighted in RED, so that they may be easily identified.

Kind regards,
Farhan Mustafa & Co-authors

**Response to Anonymous Referee 2 Comments**

**1. General Comments**

**Point 1:** While the study is within the scope of AMT, it is extremely similar to paper by Yang et al. (2019) as also pointed out by the first reviewer. The authors uses the same method but applies it to OCO-2 instead of to GOSAT data and additional apply the method to West Asia. The method section is partly copying and partly paraphrasing Section 2.3 of Yang et al. (2019). Figures 1 and 2 are also extremely similar to Figures 2 and 3 in that paper without proper citations. Equations 2 and 3 are also identical. The results and conclusions sections have also some similarities with Yang et al. (2019) in the choice of analyses and figures. It is clearly necessary to rework the method and results section to make it better understandable as well as reduce similarities with and give proper credit to Yang et al. (2009). In addition, the authors need to clarify the novelty of their paper in comparison to previous studies.

**Response 1:** We are thankful to the anonymous referee for his/her constructive comments. The comments are very helpful in improving the quality of the manuscript and we have carefully used them to revise the manuscript.

We understand the concerns of the learned referee about the similarities between our manuscript and the article authored by (Yang et al., 2019). We extended the study following the suggestion given in the conclusion of the article written by (Yang et al., 2019). However, following the suggestion of the respected reviewer, the manuscript has been revised completely and substaintial changes have been made in the revised version of the manuscript.

- The prediction model has been changed/improved. A new dataset, MODIS net primary productivity (NPP) has been added to train the model and then predict the anthropogenic $CO_2$ emission. The new flowchart of the model, updated in the revised manusctipt as, "Figure 1" is given in the following:

[Figure]

- More detail has been added to the section 2.2 of the revised manuscript at

L196-203 as, "OCO-2 XCO$_2$ dataset was downloaded from the Earthdata platform (https://earthdata.nasa.gov/) and to ensure the reliability of the data, screening and filtering of the dataset was carried out following the instructions given in the OCO-2 Data User Guide (DUG). Each sounding that is processed using the ACOS L2FP retrieval algorithm is assigned either a "good" (=0) or "bad" (=1) quality flag based on screening criteria derived from comparisons with TCCON and modelled CO$_2$ fields. It is generally advised that users should use the "good" quality soundings for regional and local scale studies because the soundings flagged as "bad" quality might include biases that compromise their utility for the application. In this study, the OCO-2 XCO$_2$ retrievals were included if: (i) they were flagged good (flag=0) and (ii) the standard deviation of the good soundings for the day was less than 2 ppm."

L245-256 as, "During the process of photosynthesis, the living plants convert the CO$_2$ into sugar molecules they use for food. In the process of making food, they also release the oxygen we breathe. Plant productivity plays a crucial role in the global carbon cycle by absorbing the CO$_2$ released by anthropogenic activities. The net primary productivity (NPP) shows how much CO$_2$ is absorbed by the plants during photosynthesis minus how much CO$_2$ is released during respiration. A negative value of NPP means that CO$_2$ is released into the atmosphere and a positive value represents the absorption of atmospheric CO$_2$. To improve the model results, an NPP dataset (MOD17A3HGF) provided by MODIS has also been used in this study. It provides information about annual NPP and is distributed by NASA's Land Processes Distributed Active Archive Center (LP DAAC). The NPP dataset with a spatial resolution of 500 meters (m) was downloaded from the LP DAAC website (https://lpdaac.usgs.gov/products/mod17a3hgfv006/). The annual NPP is derived from the sum of all 8-day Net Photosynthesis (PSN) products (MOD17A2H) from the given year. The MODIS NPP dataset was reprojected and resampled to the spatial resolution of 1°×1° Longitude/Latitude for each year and used along with the ODIAC and OCO-2 datasets to train the GRNN model and as well predicting the CO$_2$ emission."

- The sentence including, "we proposed a new method has been revised" and the author Yang et al., (2019) is given proper credit at various places of the manuscript. Such as,

At L105-107 as, "In this study, we have improved the model initially developed by (Yang et al., 2019) to estimate the regional scale anthropogenic $CO_2$ emissions using OCO-2 $XCO_2$ retrievals over East and West Asia. MODIS NPP, OCO-2 and ODIAC $CO_2$ datasets were obtained for a period of five years from January 2015 to December 2019."

At L381-388 as, "(Yang et al., 2019) estimated the $CO_2$ emissions by a similar machine learning approach using GOSAT $XCO_2$ retrievals over China and differences between the estimated and the ODIAC $CO_2$ emissions were between $-5x10^9$ kg to $5x10^9$ kg. Moreover, the predicted results from the referenced study exhibited overall less $CO_2$ emissions relative to the ODIAC emissions contradicting our results. Our study showed better results and it might be due to several reasons; (i) we improved the prediction model with the addition of NPP dataset (Figure 4e), (ii) we utilized the higher resolution XCO2 retrievals provided by OCO-2, and (iii) we incorporated the OCO-2 $XCO_2$ retrievals processed using the latest version of the retrieval algorithm. The newer version of the ACOS L2FP retrieval algorithm has improved the quantity as well as the quality of the satellite-based observations (Taylor et al., 2021)."

- Figure 1 (given above) has been changed as the model has been changed with the addition of new dataset.
- Figure 2 is a general structure of GRNN, however, the figure has been properly cited.
- The results section has been changed:

At L350-388 as, "The predicted $CO_2$ emission is overestimated over most of the regional parts; whereas, this overestimation is more significant over agricultural areas which are located near the high-density region, i.e., eastern China. Eastern China, Japan, and Korea are known to be among the regions with the highest $CO_2$ emissions and this underestimation over the agricultural areas might be caused by the nearby $CO_2$ emitting sources which raise the $CO_2$ concentration of the nearby areas through atmospheric transport. Previous studies demonstrated that the concentration of atmospheric $CO_2$ was influenced by atmospheric transport (Cao et al., 2017; Kumar et al., 2014). The areas where the predicted $CO_2$ emission is underestimated are covered by agriculture, forest and vegetation. This underestimation of the predicted $CO_2$ emissions over these areas indicate the presence of uncertainties in the $XCO_2$ anomalies that are likely to be produced by the $CO_2$ uptake of the biosphere which is still remaining in the $XCO_2$ anomalies. In addition, the areas where the estimated $CO_2$ emissions are overestimated have higher elevations. OCO-2 observations show larger uncertainties over elevated and mountainous areas, especially the Tibetan Plateau where the OCO-2 retrievals are significantly overestimated (Kong et al., 2019; Mustafa et al., 2020) and this might also have a contribution to the overestimation of estimated $CO_2$ emissions. The difference between the estimated and the ODIAC $CO_2$ emissions was ranging from $-0.06x10^9$ kg to $3.2x10^9$ kg and the magnitude of difference between $-1x10^9$ kg to $1x10^9$ kg accounted for 84% of the total number of grid cell. (Yang et al., 2019) estimated the $CO_2$ emissions by a similar machine learning approach using GOSAT $XCO_2$ retrievals over China and differences between the estimated and the ODIAC $CO_2$ emissions were between $-5x10^9$ kg to $5x10^9$ kg. Moreover, the predicted results from the referenced study exhibited overall less $CO_2$ emissions relative to the ODIAC emissions contradicting our results. Our study showed better results and it might be due to several reasons; (i) we improved the prediction model with the addition of NPP dataset (Figure 4e), (ii) we utilized the higher resolution XCO2 retrievals provided by OCO-2, and (iii) we incorporated the OCO-2 $XCO_2$ retrievals processed using the latest version of the retrieval

algorithm. The newer version of the ACOS L2FP retrieval algorithm has improved the quantity as well as the quality of the satellite-based observations (Taylor et al., 2021)."

**Point 2:** I am also not convinced by the objective of the study: What is the advantage of the suggested approach over using the ODIAC inventory for 2019? The satellite-based product seems to be less accurate suffering from issues with XCO2 accuracy, not-accounted transport effects, and biospheric fluxes. In addition, a main objective of top-down emission estimates is the evaluation/validation of bottom-up inventories, but since the GRNN is trained with the ODIAC inventory, it is not able to identify systematic errors in the ODIAC dataset. I think it will be necessary to discuss these points in the paper.

**Response 2:** The suggested discussion has been added in the revised manuscript:

At L49-60 as, "Over the past few decades, significant work has been carried out to compile the regional as well as the global inventories of $CO_2$ emission from anthropogenic activities (Olivier et al., 2005; Janssens-Maenhout et al., 2015; Gurney et al., 2009; Oda and Maksyutov, 2015). Most of the emission inventories employ 'bottom-up' methods using available human activity data, emission factors and corresponding technologies. The bottom-up methods incorporate energy consumption datasets along with other information such as fuel purity, efficiency, etc. However, it is known that such information can be subject to errors and biases leading to considerable discrepancies and uncertainties in emission estimates, especially in the case of rapidly growing developing economies such as China and India (Guan et al., 2012; Korsbakken et al., 2016). These discrepancies can result in ~40% to ~100% uncertainty in emission estimations at the country and the local scales, respectively (Peylin et al., 2013; Wang et al., 2013). Moreover, the uncertainty in inventory datasets is also a challenging task and the intercomparisons of various inventories do not necessarily reveal all the uncertainties because different inventories are sometimes using common sources of information (Konovalov et al., 2016)."

At L111-117 as, "Atmospheric $CO_2$ monitoring satellites can detect and analyze the anthropogenic $CO_2$ signatures and the satellite-based estimation of anthropogenic $CO_2$ emissions can be helpful in investigating the carbon emissions as a data-driven method, which is different to the conventional method in calculating emission inventory. Although estimation of anthropogenic $CO_2$ emission using satellite datasets is a challenging task because some other factors such as the atmospheric transport and the terrestrial ecosystem play notable roles in controlling the spatial distribution of atmospheric $CO_2$ (Cao et al., 2017) but still this data-driven method can provide a meaningful help in quantifying anthropogenic $CO_2$ emissions that will be important for evaluating the effects for anthropogenic $CO_2$ emissions reduction at regional as well as global scales."

**2. SSpecific/technical comments**

L72: Please clarify that this is not a new method.
**Response :** It has been cleared in the revised manuscript:
At 105-107 as, "In this study, we have improved the model initially developed by (Yang et al., 2019) to estimate the regional scale anthropogenic $CO_2$ emissions using OCO-2 XCO$_2$ retrievals over East and West Asia."

L111: life period -> life time
**Response :** The mistake is corrected.

L116ff: The paragraph is unclear. Please describe more clearly how the XCO2 anomaly is calculated.
**Response :** We are thankful to the reviewer for valuable suggestion. The description has been simplied in the revised manuscript:
At L208-210 as, "To highlight the areas associated with the anthropogenic $CO_2$ emission, $XCO_2$ anomalies were calculated by subtracting the daily $XCO_2$ median (daily background) from the individual $XCO_2$ observation, a method suggested by previous studies (Hakkarainen et al., 2019, 2016)."

L175: Contributing a large fraction of global oil production does not necessarily imply high CO2 emissions.
**Response :** The sentence has been removed.

L176: "major fuel consumer" compared to whom?
**Response :** Compared to other countries in the region. The sentence has been revised as, "In addition, Iran, Saudi Arabia, and Iraq are the major fuel consumers of the region and contribute more than 60% of the region's total fossil fuel $CO_2$ emissions."

L179: "highest" compared to whom? Maybe just write "high" here?
**Response :** The mistake has been corrected.

L201: The term "actual emissions" might refer to "true emissions", which are unknown. I would suggest to use "ODIAC inventory" here.
**Response :** We are thankful to the reviewer for valuable suggestion. The term "actual emission" has been revised throughout the manuscript.

L213f: It quite unclear what the "difference" and "magnitude of difference" refer to. Instead of stating exponential values here, it would more interesting what are the absolute and relative deviations depending, for example, on land cover.
**Response :** We are grateful to the learned reviewer for constructive comment. The manuscript has been revised following the given suggestion.

L214: What does "accounted for 80% of the total grids" mean?
**Response :** It mean the 80% of the toal number of grid cells. The sentence has been simplied as, "the magnitude of difference between $-1x10^9$ kg to $1x10^9$ kg accounted for 84% of the total number of grid cells."

L215f: When comparing to Yang et al. (2019) it would useful using the same units.
**Response :** Same units have been used for comparions in the revised manscript:
At L379-388 as, "The difference between the estimated and the ODIAC $CO_2$ emissions was ranging from $-0.06x10^9$ kg to $3.2x10^9$ kg and the magnitude of difference between $-1x10^9$ kg to $1x10^9$ kg accounted for 84% of the total number of grid cells. (Yang et al., 2019) estimated the $CO_2$ emissions by a similar machine learning approach using GOSAT $XCO_2$ retrievals over China and differences between the estimated and the ODIAC $CO_2$ emissions were between $-5x10^9$ kg to $5x10^9$ kg. Moreover, the predicted results from the referenced study exhibited overall less $CO_2$ emissions relative to the ODIAC emissions contradicting our results. Our

study showed better results and it might be due to several reasons; (i) we improved the prediction model with the addition of NPP dataset (Figure 4e), (ii) we utilized the higher resolution XCO2 retrievals provided by OCO-2, and (iii) we incorporated the OCO-2 $XCO_2$ retrievals processed using the latest version of the retrieval algorithm. The newer version of the ACOS L2FP retrieval algorithm has improved the quantity as well as the quality of the satellite-based observations (Taylor et al., 2021)."

L239: "A previous study…" -> "Yang et al. (2019) …"
**Response :** Change has been made in the revised manuscript.

L236ff: Figure 6b shows some clear deviation from the linear relationship. Do you have an explanation for this behavior?
**Response :** This behaviour is due to the reason that XCO2 anomalies show strong correlation with the higher values of ODIAC emissions, however, this correlation is weak with the smaller values of ODIAC inventory.

L259: Please clarify that this approach was suggested already by Yang et al. (2019).
**Response :** It has been clarified in the Introduction section and this misleading sentence in the Summary and Conclusions section has been removed in the revised manuscript.

L275ff: You could mention some current and future satellites here that could be used to improve the approach.
**Response :** The suggestion has been improved and the manuscript has been revised:
At L630-631 as, "Joint utilization of the observations from the old and the latest satellites such as OCO-3, GOSAT-2, and TanSAT might reduce the spatiotemporal gaps and uncertainties."

Figs. 3-6: You use blue or bright colors for high emissions and red or dark colors for low emissions, which is somewhat counterintuitive because most people would expect the opposite.
**Response :** We are thankful to the learned reviewer for constructive comment. We have updated the maps following the colour pallets used by most of the CO2 community, i.e., Perceptually Uniform Sequential color pallete, "Viridis" for emissions and diverging colour pallete "RdYlBu" for maps showing differences.

[Figure]

**References**

[revised manuscript text omitted]